

# Intensive sea urchin harvest rescales *Paracentrotus lividus* population structure and threatens self-sustenance

Nicole Ruberti[1], Gianni Brundu[2], Giulia Ceccherelli[3,6], Daniele Grech[2], Ivan Guala[2], Barbara Loi[2] and Simone Farina[4,5,6]

[1] Department of Architecture, Design and Planning, University of Sassari, Sassari, Italy
[2] IMC-International Marine Centre, Torre Grande (OR), Italy
[3] Department of Chemical Physical Mathematical and Natural Science, University of Sassari, Sassari, Italy
[4] Department of Integrative Marine Ecology (EMI), Stazione Zoologica Anton Dohrn–National Institute of Marine Biology, Ecology and Biotechnology, Genoa Marine Centre, Genoa, Italy
[5] National Research Council, Institute for the study of Anthropic Impacts and Sustainability in the Marine Environment (CNR-IAS), Torre Grande, Italy
[6] NBFC, National Biodiversity Future Center, Palermo, Italy

Corresponding author
Simone Farina, simone.farina@szn.it

## ABSTRACT

The harvest of the edible sea urchin *Paracentrotus lividus* is intensively practiced in some regions of the Western Mediterranean Sea. The removal of the largest individuals can determine an overall reduction in population size and a size class truncation that can lead to a drastic drop the self-sustenance. The aim of this study is to evaluate the variability of the population reproductive potential across 5 years in one of the main harvest hotspots of Sardinia (Western Mediterranean Sea). The breeding stock consists of commercial and under-commercial size individuals which were sampled on a monthly basis to estimate their GonadoSomatic Index (GSI) and the Individual Gamete Output (IGO). In addition, the reproductive potential of the population—Total Gamete Output (TGO)—was calculated across the 5-year period in relation with the variation of the density of the breeding stock. During the last year, the reproductive potential was also estimated in a well-conserved population of a nearby Marine Protected Area. No significant variability in GSI and IGO was found over the 5 years nor when compared with the ones of protected population in the last year. However, the intensive harvest drastically rescaled the population body-size: although density of the commercial size class remained low, density of the under-commercial size-class halved from the beginning to the end of the study. Accordingly, the proportional decrease of their gamete output contribution led to a 40% loss of the reproductive potential of the whole population in the 5-year period. Interestingly, despite the loss of reproductive potential due to the decrease of the breeding stock density, the average values of IGO slightly increased across the years leading to the highest Annual Gamete Output (AGO) during the fourth year of sampling. This positive pattern could suggest a mechanism of reproductive investments of the survivors in terms of gonad production rate or increase in spawning intensity. This work provides evidence of the direct effect of size-selective harvesting on the rapid loss of population self-sustenance. Furthermore, it lays new prospective for future research of the indirect effects of the rescaling population body-size in functional traits of the sea urchin

*P. lividus* and that could become important for both, sustainable exploitation and ecosystem conservation management.

## INTRODUCTION

Fishing is the most widespread human exploitative activity in the marine environment, and it is size-selective by definition (*Longhurst, 2006*). Worldwide well-managed fisheries are based on removing individuals above a minimum legal size (*Jackson et al., 2001*) and in theory juveniles can grow large enough to reproduce at least once before being harvested, guaranteeing a sustainable harvest (*Law, 2000*). However, nowadays fishing is so intense (*Baum et al., 2003*; *Myers & Worm, 2003*; *Pauly et al., 1998*) that affects many aspects of the biology of a target species, such as demography, life history and ecology (*Fenberg & Roy, 2008*).

One of the main effects of the size-selective harvest is the deep change in the demographic structure of the population. Specifically, the continuous removal of the largest individuals can determine an overall reduction in population abundance and body-size structure (*Fenberg & Roy, 2008*). Thus, under heavy size-selective harvest, an age-size truncation of the populations occurs, leading to a multitude of consequences (*Festa-Bianchet, 2003*; *Heino & Godo, 2002*). Firstly, this phenomenon is critical for the self-sustainability of populations and involves the loss of the main breeding stock and a serious decline of population density (*Enberg, Jørgensen & Mangel, 2010*). Effectively, it is widely demonstrated that large individuals give the greatest contribution to the successful offspring and the larval size and quality of some exploited marine fish have been shown to be positively correlated to maternal length and age (*Berkeley, Chapman & Sogard, 2004*; *Trippel, 1995*; *Vallin & Nissling, 2000*). Thus, the removal of largest and oldest individuals generally decreases the population ability to replenish itself.

The animal body-size is central to ecology, from the organismal physiology to the functioning of communities and ecosystems (*Peters, 1983*). The intensive size-selective harvest can cause alterations in the growth rate and in the timing of maturation of youngest specimens (*Hamilton et al., 2007*). The decrease in mean size and abundance of the target fishes can also generate negative effects on non-target species through the food web interactions (*Audzijonyte et al., 2013*), for example reducing the predator-prey interactions and causing the proliferation of preys (*e.g.*, *Pinnegar et al., 2000*). This mechanism is widely demonstrated in temperate reef tri-trophic interactions between fish, sea urchins and macrophytes. Overfishing depletes populations of predatory fish and the substantial loss of large predators that exert a top-down control mechanism, causing an unregulated increase in sea urchin population densities (*Guidetti, Boero & Bussotti, 2005*; *McClanahan & Shafir, 1990*; *Micheli et al., 2005*; *Sala et al., 2012*; *Sala, Boudouresque & Harmelin-Vivien, 1998*; *Shears & Babcock, 2003*).

The indirect impact of the size-selective harvesting on the ecosystems largely depends on the functional role and competitive dominance of the target species (*Kaiser & Jennings, 2001*). For example, heavy size-selective harvest of sea urchin top-predators generates cascading effects pushing the system beyond the resilience tipping points (*Ling et al., 2015*), in extreme cases facilitating the shift from vegetated coastal marine ecosystems to bare rocky areas—barrens—hosting low biodiversity (*e.g.*, *Bianchelli & Danovaro, 2020*).

In the last decades the lower-trophic-level fisheries intensified the exploitation on remaining commercial species, including invertebrates (*e.g.*, *Anderson et al., 2011*), among which sea urchins represent a relevant economic resource (*Andrew et al., 2002*). Sea urchin fisheries further worsen the functioning of the ecological system: in fact, in addition to the effects due to low density and biomass of top-predators due to overfishing, the intensive sea urchin harvest leads to their abrupt decline and, ultimately, determining the collapse of their populations (*e.g.*, *Johnson et al., 2012*).

Since the gonad size is proportional to the sea urchin size, being more mature and developed in the largest individuals (*Mita et al., 2007*), the size-selective harvest of the largest sea urchins can gradually compromise the population fertility (*Brewin et al., 2000*; *Byrne, 1990*; *Guettaf, San Martin & Francour, 2000*). Populations under heavy harvest pressure are often destined to collapse unless they receive enough larval supply from the outside (*Dubois et al., 2016*), determining important changes in the community structure (*Steneck et al., 2002*; *Vadas & Steneck, 1995*; *Villouta, 2000*).

The edible sea urchin *Paracentrotus lividus* (Lamarck 1816) is the most exploited species in the Mediterranean Sea but, at the same time, it is an indispensable functional herbivore in controlling macroalgal communities through their grazing activity (*Boada et al., 2017*; *Guarnieri et al., 2014*; *McClanahan & Sala, 1997*; *Prado et al., 2007*). In some regions of the Western Mediterranean Sea, the intensive harvest of *P. lividus* locally re-scaled population body-size determining an evident cut-off of their population structure above the commercial size-class of 50 mm (test diameter without spines; TD) and with the depletion of the breeding stock (*Couvray et al., 2015*; *Gianguzza et al., 2006*; *Guidetti, Terlizzi & Boero, 2004*).

In Sardinia (Italy, Western Mediterranean Sea), *P. lividus* is extensively harvested since the early 2000s which has dramatically reduced the sea urchin density in extensive coastal areas (*Ceccherelli et al., 2011*; *Pais et al., 2012*, *2007*; *Pieraccini, Coppa & de Lucia, 2016*; *Ceccherelli et al., 2022*). In the worst cases, the loss of the largest and oldest individuals shifted the reproductive potential of the population to the smaller fertile individuals (*Loi et al., 2017*).

Estimating through time the gamete production of the exploited sea urchin populations can be crucial to manage the local fishery sustainability and to prevent the population collapse. The spatial and temporal variation of the gamete production can be assessed through the analysis of the annual reproductive cycle (*Brewin et al., 2000*).
The reproductive cycle is generally evaluated through the estimation of the GonadoSomatic Index (GSI), as fluctuations in gonad size and spawning periods (*Gianguzza et al., 2013*; *Shpigel et al., 2004*; *Spirlet, Grosjean & Jangoux, 1998*), and from which the reproductive potential of a population (the total gamete output) depends on

(*Brewin et al., 2000*). The variation of GSI is strictly associated to environmental changes, such as sea surface temperature (*Beddingfield & McClintock, 1998*; *Levitan, 1991*; *Levitan, Sewell & Chia, 1992*; *Spirlet, Grosjean & Jangoux, 2000*; *Spirlet, Grosjean & Jangoux, 1998*), wave height and food quality (*Brady & Scheibling, 2006*; *Byrne, 1990*; *Gianguzza et al., 2013*; *Lozano et al., 1995*; *Minor & Scheibling, 1997*; *O'Hara & Thórarinsdóttir, 2021*; *Sellem & Guillou, 2007*).

Overall, the largest adults represent the major breeding stock of the population but at the same time are the target commercial size-class (TD ≥ 50 mm), whilst the smaller fertile individuals at under-commercial size-class (40 ≤ TD < 50 mm), are important contributors for the self-sustenance of the population (*e.g.*, *Loi et al., 2017*) and represent the future fishing stock. The breeding stock can produce one or more cohort of mature gametes in a single breeding season (*Mita et al., 2007*), being their reproductive cycle characterized by one or more seasonal peaks (*Boudouresque & Verlaque, 2001*; *Ouréns, Fernández & Freire, 2011*).

The aim of this study is to determine the variability across 5 years of the reproductive potential in one of the most harvested sea urchin populations of Sardinia. The reproductive potential was evaluated by the estimation of GSI and IGO in relation to the population density. During the last year of the study, a comparison of the reproductive potential of an unexploited population (in the nearby Marine Protected Area) was also done.

The reproductive potential corresponds to the gamete output produced by the whole population in relation with the portion of the breeding stock (large and small fertile individuals). In the study area the annual trend of the Gonadosomatic Index and the consequently gamete output were expected not to change over the years, with the exception of the occurrence of sea surface temperature or wave height anomalies. Conversely, the intense size-selective harvest that systematically removes the largest breeder contributors, would be the cause of an important decrease in the reproductive potential of the population. Accordingly, a drastic decline of the reproductive potential due to the rescaling population body-size is supposed.

## MATERIAL AND METHODS

### Study site

This study was conducted at Su Pallosu Bay (40.0489°N, 8.4161°E), located in the north of the Peninsula of Sinis (central western Sardinia), a high natural density area of sea urchins that has been overexploited by fishermen for many years (*e.g.*, *Coppa et al., 2021*; *Fois et al., 2020*; *Farina et al., 2020*).

The favorable environmental conditions, such as the shallow calcareous plateau with *Posidonia oceanica* patches (*De Falco et al., 2008*) support a high sea urchin density in this area. The low current speed determining recirculation cells of water surface at Su Pallosu Bay promotes the recruitment success of the population respect to the nearby Marine Protected Area (*Farina et al., 2018*). The low abundance of top-predators (*Marra et al., 2016*), typical of the fishing areas, reduces the natural mortality of juveniles and medium size-classes (*Oliva et al., 2016*; *Farina et al., 2020*).

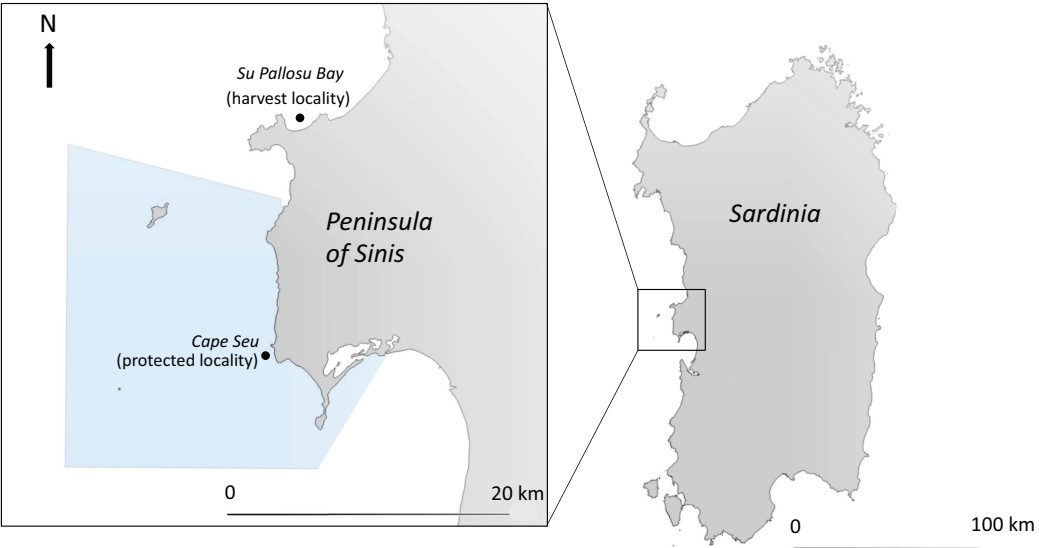

**Figure 1 Map of the study area.** In the North of Sinis Peninsula, Su Pallosu Bay is one of the most important harvest hot-spot in West coast of Sardinia (harvest locality), while Cape Seu is a locality within the nearby Marine Protected Area (area in light blue) and monitored during the last period (protected locality). Sea urchin sampling sites are represented by the black dots.

Accordingly, Su Pallosu Bay is considered one of the most important sea urchin harvest hotspots in Sardinia (Fig. 1). In this area, as well as in the whole island, the professional sea urchin harvest is officially open from November to April and it is allowed with scuba diving. Since 2009, 189 professional licenses have been released and the limit of the daily catch amount has been fixed to 1,500 per fisherman or 3,000 per boat, while the minimum urchin catch size (test dimeter with no spines) is 5 cm (RAS, Autonomous Region of Sardinia, decree no. 2524/DecA/102 of 7 October 2009).

In the nearby Marine Protected Area ("Penisola del Sinis—Isola di Mal di Ventre", established in 1997; Fig. 1) the sea urchin harvest was intensively practiced before 2009 (*Coppa et al., 2021*). Since 2009 this activity progressively decreased and, nowadays, it is restricted in terms of fishing licenses (55 licenses are allowed only for residents in this area), of catch quota per day which amounts to 500 individuals, while recreational fishing has been banned (*Farina et al., 2020*; *Ceccherelli et al., 2022*). Inside the MPA, the reproductive potential of the unexploited sea urchin population of Cape Seu (39.8980°N, 8.4010°E) was sampled during the last year of study.

The harvest and protected localities have homogeneous macroalgal communities (*Guala et al., 2006*; *Anedda et al., 2021*) and are far from urbanizations, harbours, aquaculture activities and rivers. Accordingly, there was no anthropogenic influence on the dissolved nutrient concentration on macroalgal composition and abundance over the years (*Loi et al., 2017*). Moreover, dispite the low abundance of top predators, no barren grounds are present in this area.

The annual average of the Sea Surface Temperature (SST) in the area ranged between 17.5 and 19 °C, with colder open waters and warmer coastal waters (*Cucco et al., 2006*).

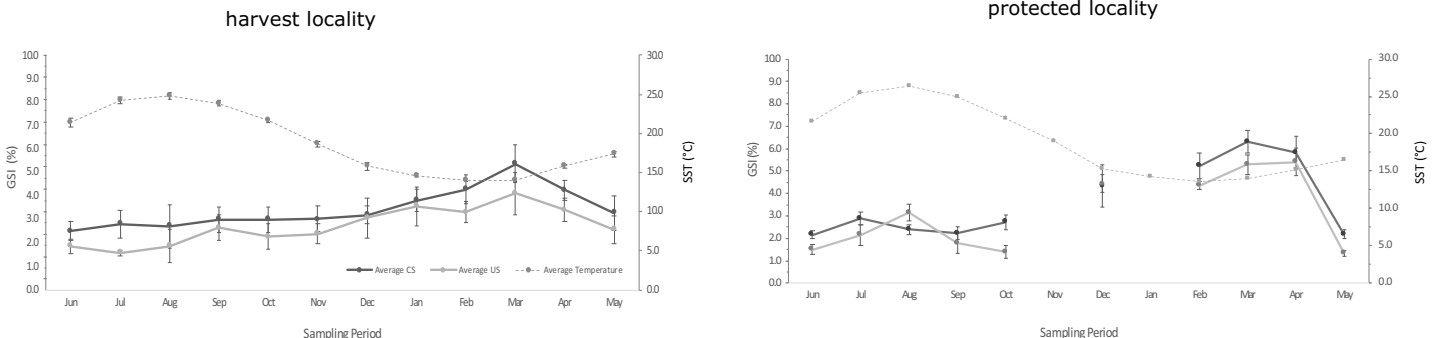

**Figure 2 Graphs of GSI over the sampling periods.** Average annual trend of the GonadoSomatic Index in the harvest locality and in the protected locality during the last sampling period (P5). GSI is represented as mean ± standard deviation for CS size-class (black line) and US size-class (gray line). Mean SST was also plotted (thinner gray line).

The wind-waves are generated on a wide fetch by the prevailing winds that, in the form of severe winter storms, can produce intense Sea Wave Height (SWH) of up to 5 m (*Simeone et al., 2016*). Data of the daily SST were extrapolated from NOAA dataset and downloaded from the "Asia-Pacific Data Research Centre" page at SOEST (https://www.soest.hawaii.edu/soestwp/research/org/centers/asia-pacific-data-research-center-apdrc/). Data of the hourly SWH (m) were downloaded from Copernicus Marine Environmental Monitoring Service (https://resources.marine.copernicus.eu/?option=com_csw&task=results&pk_vid=20be7e57367a57a51616599440425bbc). The monthly mean SST and SWH were calculated choosing an intermediate point between the two sampling sites, representing the trend of SST and SWH in each locality during the sampling period.

## GonadoSomatic index and annual gamete output

Sea urchins were monthly sampled across 5 years from 2013 to 2019 at the harvest locality and in 2018-19 at the protected locality. However, due to adverse marine weather conditions persisting over time, samplings could not be done in a few months (Figs. 2, S3).

The sea urchins were collected by scuba diving at two sites few hundred meters apart and over a rocky bottom at the bathymetry of 5 ± 1 m, corresponding to the mean depth at which the harvesters usually work. During monitorings, 12 to 20 individuals were collected for each site, both for the commercial (CS, test diameter TD ≥ 50 mm) and the under-commercial size-class (US, 40 ≤ TD < 50 mm).

In accordance with maturity rating proposed by *Byrne (1990)*, sea urchins were in recovery stage mainly during summer and autumn, from June to October. Gametogenesis began at the end of autumn and lasted until early spring, fully mature stage was found from January to April and spawning events occurred mostly in winter-spring (*Ghisaura et al., 2016*; *Siliani et al., 2016*). In general, there are no significant asynchronisms in the reproductive cycle between males and females of *P. lividus* (*Crapp & Willis, 1975*) and 1:1 sex ratio was generally observed in this specific populations (*e.g.*, *Loi, 2018*). Finally, population fecundity in this area can be considered suitable accordingly to the analysis conducted by *Loi et al. (2017)* and the eggs fertilization rate obtained in *Brundu et al. (2016, 2017)*.

The gonadosomatic index (GSI) represents the gonad mass as a proportion of the total body mass of the animals (*Lawrence, Lawrence & Holland, 1965*). Accordingly, size (cm of test diameter without spine) and wet weight (g) were recorded in each sea urchin, gonads collected and weighted (g), and the GSI of each sampled individual calculated as:

$$([gonad\ wet\ weight/total\ wet\ weight] \times 100)\%$$

(*Lawrence, Lawrence & Holland, 1965*)

Accordingly, the individual GSI were pooled together by month to determine monthly mean GSI by size classes by locality (monthly mean GSI; *Brewin et al., 2000*). The complete annual reproductive cycle, described by GSI calculated from May of 1 year to June of the following year, represents a sampling period. Thus, GSI of both commercial and under-commercial, size-classes were monthly calculated for five sampling periods (P1–P5) at the harvest locality and for the last sampling period (P5) at the protected locality.

The GSI annual pattern was estimated for the whole population as the average of the monthly mean GSI values for each sampling period in each locality. The highest and the lowest peaks of the monthly mean GSI recorded over a sampling period correspond to the time before the beginning (pre-spawning) and after the end (post-spawning) of a spawning event.

The mean individual gamete output (IGO) was then calculated as:

$$(pre\ spawning\ GSI\ -\ post\ spawning\ GSI)/100\ g\ g^{-1}se^{-1}$$

(*Brewin et al., 2000*)

Which is the difference between the monthly mean pre-spawning GSI and the monthly mean post-spawning GSI in units of gamete wet weight per sea urchin per spawning event ($g\ g^{-1}\ se^{-1}$; *Brewin et al., 2000*). The sum of the all-year-round differences (IGOs) represents the Annual Gamete Output (AGO), which is the reproductive contribution per year of each fertile size-class and it is measured in units of gonad wet weight per sea urchin wet weight per year ($g\ g^{-1}\ yr^{-1}$; *Brewin et al., 2000*).

## Population density and reproductive potential

Sea urchin population structure was estimated during P1 (2013-14) and P5 (2018-19) sampling periods at the harvest locality, and during P5 (2018-19) at the protected locality as well. Density and size frequency distribution of sea urchins of both localities were evaluated in the same two sites where GSI was estimated by independent underwater counts (*Guala et al., 2006*). The counts were carried out in three replicates of 5m² (20 contiguous 50 × 50 cm quadrats) each, the minimum optimal surface to detect aggregative distribution of sea urchins, as consequence of the habitat heterogeneity.

All the animals found in the sampling quadrats were counted and measured with a calliper. The abundance was then estimated as density (ind m⁻²) and individuals were grouped in size-classes of 10 mm of test diameter to build the population structure (*e.g.*, *Farina et al., 2020*, *2022*).

Finally, the population reproductive potential was estimated as the total gamete output (TGO) for each fertile size-class and for the whole population per m² per year (popTGO)

and defined as the sum of the AGO of each fertile size-class multiplied for the sea urchin density (g g$^{-1}$ m$^{-2}$ yr$^{-1}$; *Loi et al., 2017*).

## Data analysis

Variability of the monthly mean SST at the harvest locality was evaluated among the sampling periods P1-P5 (from 2013 to 2019) throughout the non-parametric analysis of variance Kruskal-Wallis test (*Kruskal & Wallis, 1952*). Moreover, the monthly mean SST during P5 was compared with the SST at the protected locality (Mann-Whitney test; *Mann & Whitney, 1947*). Similarly, difference in the monthly mean SWH was estimated among the sampling periods with a parametric analysis of variance (Anova 1-way, *Underwood, 1997*). Since the SST and SWH strongly influence the annual trend of GSI (see the introduction), in order to exclude their statistical effects, they were successively set as further predictors with random distribution and independent from the response variables in the following analysis.

Exploration of GSI and AGO data was carried out to check that ANOVA assumptions (*i.e.*, normal distribution and homogeneity of variance) were met (*Zuur, Ieno & Elphick, 2010*). Since the GSI data were characterized by a different number of replicates collected during the years, and non-normal distribution, General linear mixed model (GLMM) with Poisson family was chosen as the best tool to handle unbalanced data involving random factors (SST and SWH).

A GLMM was performed to assess the variance of the monthly mean GSI in the harvest locality setting "Period" (five levels) and "Size" (two levels) as fixed factors, "SST" (twelve levels) and "SWH" (twelve levels) nested "Period" and "Site" (two levels) as random factors in order to exclude their statistical effects on the response variable. A second GLMM was run to evaluate differences in the monthly mean GSI between harvest and protected localities during P5. This analysis was set with "Locality" (two levels) and "Size" (two levels) as fixed factors and, "SST", "SWH" and "Site" as random factors. A linear mixed model (LMM) was also performed to investigate potential significant differences in AGO emitted by the harvested population over the sampling periods. The model was set with AGO as response variable, "Period" as fixed factor, and "Size" as random factor. Graphical validations of all the statistical models are provided in the Supplementary Materials.

Finally, a statistical analysis to evaluate differences in the density of sea urchin population structures was carried out between P1 and P5 in the harvest locality. LMMs were performed setting density of recruits (TD < 10 mm), juveniles and middle size-classes (10 ≤ TD < 40 mm), density of US and CS size-classes as response variables and "Period" and "Site" as fixed and random factors, respectively.

All the described procedures and analyses were performed using the Nortest and lme4 packages in R software (*R Core Team, 2010*).

## RESULTS

The monthly mean SST did not change significantly neither in the harvest locality through time (Kruskal Wallis test: chi-squared = 0.43005, *p*-value = 0.98; Fig. 2 and Table S1), nor

**Table 1 GLMM results for GonadoSomatic index in function of (a) "Period" and "Size" across the 5-year period and, (b) "Locality" and "Size" during the last sampling period P5.**

| (a) | Fixed effects | Estimate coeff | Std. error | z-value | p-value |
|---|---|---|---|---|---|
| GSI | Intercept | 1.14816 | 0.11983 | 9.582 | $<2e^{-16}$ |
| | Period 2 | 0.13448 | 0.15556 | 0.864 | 0.387 |
| | Period 3 | 0.23617 | 0.15456 | 1.528 | 0.127 |
| | Period 4 | −0.12932 | 0.16203 | −0.798 | 0.425 |
| | Period 5 | −0.25762 | 0.16261 | −1.584 | 0.113 |
| | Size | −0.25406 | 0.03251 | −7.815 | **$5.5e^{-15}$** |
| | **Random effect** | **Variance** | **Std. dev** | | |
| | Site | $4.517e^{-03}$ | 0.067211 | | |
| | Period:SST | $1.259e^{-01}$ | 0.354813 | | |
| | Period:SWH | $1.225e^{-09}$ | 0.000035 | | |

AIC = 5,903.2; $R^2$ = 0.39

| (b) | Fixed effects | Estimate coeff | Std. error | z-value | p-value |
|---|---|---|---|---|---|
| GSI | Intercept | 3.4725 | 0.4812 | 7.217 | $5.42e^{-06}$ |
| | Locality | −0.8230 | 0.4084 | −2.015 | 0.0695 |
| | Size | −0.4714 | 0.1196 | −3.943 | **$9.03e^{-05}$** |
| | **Random effect** | **Variance** | **Std .Dev** | | |
| | Site | 0.01395 | 0.1181 | | |
| | SST | 0.76585 | 0.8751 | | |
| | SWH | 1.23354 | 1.1106 | | |

AIC = 2,199.4; $R^2$ = 0.54

**Note:**
The "SST", "SWH" and "Site" are set as random effects. Estimate coefficient, Standard Error, z-value and significance level (p-value) are provided for fixed effects. Values in bold are statistically significant.

between the harvest and the protected localities during P5 (Mann-Whitney test: p-value = 0.7561; see Fig. 2 and Table S1). Furthermore, the difference in the monthly mean SWH was not significant across the 5-year period (Anova one-way test: p-value = 0.579; Table S2).

## GonadoSomatic Index and annual gamete output

Overall, an annual average of 400 sea urchins of both fertile size-classes (CS and US) were collected during the 5-year period to estimate the monthly mean GSI trend in the harvest locality (Fig. S1). The monthly mean GSI was significantly influenced by the "Size" (p-value < 0.001), but not by the "Period" (AIC = 5903.2, $R^2$ = 0.39; see Fig. 2, Table 1A and Fig. S2). The monthly mean GSI compared between localities (the harvest and the protected) during P5 was also significantly influenced by the "Size" (p-value = 0.0019), but not by the "Locality" (AIC = 2199.4; $R^2$ = 0.54; see Fig. 2, Table 1B and Fig. S3).

The main pre-spawning GSI was in March, except for the P3 that was in August, while smaller spawning events were observed in Autumn (Fig. 2, Table 2 and Fig. S1). The main pre-spawning GSI ranged between 5.5 ± 0.71% during P2 and 7.07 ± 0.43% during P5 for CS, while US ranged between 4.05 ± 0.52% and 5.48 ± 0.42% during P3 and P5, respectively (Fig. 2 and Table 2). In the protected locality the main pre-spawning GSI was

**Table 2 Summary table of GonadoSomatic index, individual gamete output and annual gamete output.** The table shows mean monthly GSI, IGO and AGO of populations in harvest locality and protected locality during the sampled periods (P1–P5). The sea urchin density and the related reproductive potential in P1 and P5 for population in the harvest locality and in P5 for the population in protected locality are also reported.

| Period | Size-class | GSI (%) pre-spawning | GSI (%) post-spawning | IGO (g g$^{-1}$ se$^{-1}$) | AGO (g g$^{-1}$ yr$^{-1}$) | Urchin density (ind m$^{-2}$) | TGO (g g$^{-1}$ m$^{-2}$ yr$^{-1}$) |
|---|---|---|---|---|---|---|---|
| 1 | CS | 2.39 ± 0.13 | 2.17 ± 0.15 | 0.002 | 0.06 | 0.6 ± 0.2 | 0.03 |
|   | CS | 2.95 ± 0.26 | 2.52 ± 0.16 | 0.004 |      |          |      |
|   | CS | 6.55 ± 0.29 | 1.61 ± 0.11 | 0.049 |      |          |      |
|   | US | 1.76 ± 0.14 | 1.14 ± 0.14 | 0.006 | 0.04 | 2.7 ± 0.3 |     |
|   | US | 2.61 ± 0.32 | 2.07 ± 0.19 | 0.005 |      |          | 0.12 |
|   | US | 4.43 ± 0.39 | 1.33 ± 0.21 | 0.031 |      |          |      |
|   | Pop |            |             | 0.016 | 0.11 | 3.3 ± 0.5 | 0.15 |
| 2 | CS | 2.72 ± 0.50 | 2.20 ± 0.19 | 0.005 | 0.03 | -        | -    |
|   | CS | 2.44 ± 0.23 | 2.25 ± 0.28 | 0.002 |      |          |      |
|   | CS | 3.67 ± 0.34 | 3.11 ± 0.31 | 0.006 |      |          |      |
|   | CS | 5.45 ± 0.61 | 4.03 ± 0.57 | 0.014 |      |          |      |
|   | US | 1.83 ± 0.27 | 1.33 ± 0.88 | 0.005 | 0.05 | -        | -    |
|   | US | 2.99 ± 0.18 | 1.53 ± 0.90 | 0.015 |      |          |      |
|   | US | 5.47 ± 0.64 | 2.73 ± 0.53 | 0.027 |      |          |      |
|   | Pop |            |             | 0.010 | 0.08 | –        | –    |
| 3 | CS | 5.33 ± 0.62 | 3.96 ± 0.40 | 0.014 | 0.03 | -        | -    |
|   | CS | 4.09 ± 0.33 | 2.46 ± 0.37 | 0.016 |      |          |      |
|   | US | 4.05 ± 0.52 | 3.26 ± 0.51 | 0.008 | 0.03 | -        | -    |
|   | US | 3.96 ± 0.47 | 2.72 ± 0.69 | 0.012 |      |          |      |
|   | US | 1.97 ± 0.33 | 0.63 ± 0.07 | 0.013 |      |          |      |
|   | Pop |            |             | 0.012 | 0.06 | –        | –    |
| 4 | CS | 4.27 ± 1.34 | 1.52 ± 0.37 | 0.027 | 0.08 | -        |      |
|   | CS | 3.25 ± 0.40 | 2.04 ± 0.26 | 0.012 |      |          | –    |
|   | CS | 5.34 ± 0.58 | 1.73 ± 0.47 | 0.036 |      |          |      |
|   | US | 4.23 ± 0.98 | 2.24 ± 0.56 | 0.020 | 0.05 | -        | –    |
|   | US | 4.59 ± 0.32 | 1.40 ± 0.19 | 0.032 |      |          |      |
|   | Pop |            |             | 0.025 | 0.13 | –        | –    |
| 5 | CS | 2.72 ± 0.34 | 0.89 ± 0.13 | 0.018 | 0.07 |          |      |
|   | CS | 1.65 ± 0.22 | 1.38 ± 0.26 | 0.003 |      | 0.5 ± 0.1 | 0.04 |
|   | CS | 7.07 ± 0.43 | 2.11 ± 0.30 | 0.050 |      |          |      |
|   | US | 1.40 ± 0.19 | 0.98 ± 0.18 | 0.004 | 0.04 | 1.2 ± 0.4 | 0.05 |
|   | US | 5.48 ± 0.42 | 1.73 ± 0.21 | 0.038 |      |          |      |
|   | Pop |            |             | 0.022 | 0.11 | 1.7 ± 0.5 | 0.09 |
| 5P | CS | 2.88 ± 0.29 | 2.22 ± 0.29 | 0.007 | 0.05 | 0.9 ± 0.4 | 0.04 |
|   | CS | 6.30 ± 0.50 | 2.17 ± 0.20 | 0.041 |      |          |      |
|   | US | 3.14 ± 0.38 | 1.39 ± 0.28 | 0.018 | 0.06 | 0.8 ± 0.4 | 0.05 |
|   | US | 5.41 ± 0.61 | 1.35 ± 0.13 | 0.041 |      |          |      |
|   | Pop |            |             | 0.026 | 0.11 | 1.7 ± 0.8 | 0.09 |

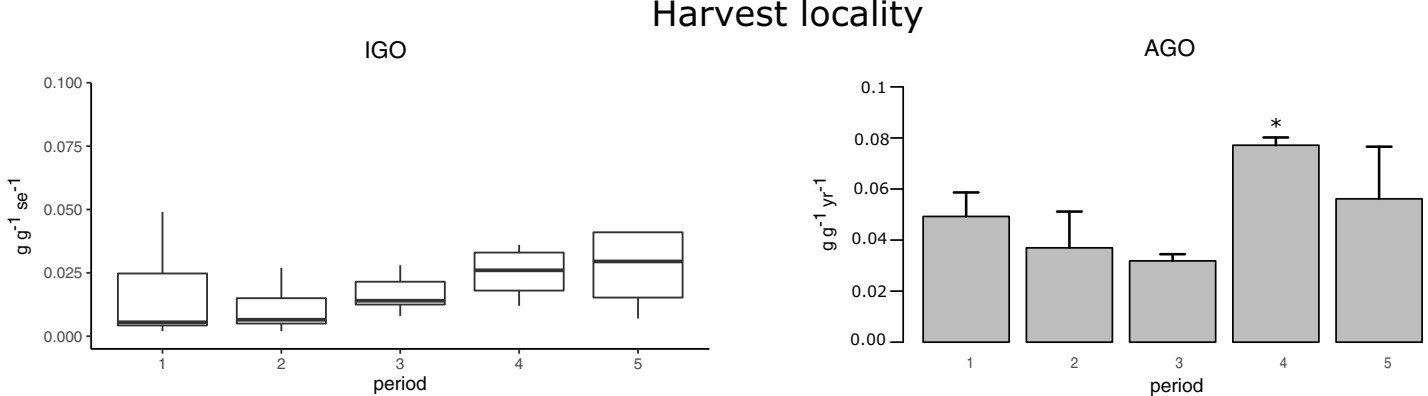

**Figure 3 Graphs of the individual gamete output and annual gamete output over the sampling periods.** The inter-annual trend of individual gamete output (IGO, g g$^{-1}$ se$^{-1}$) and annual gamete output (AGO, g g$^{-1}$ yr$^{-1}$) in the harvest population over the sampling periods. An asterisk (*) indicates significant difference.

**Table 3 LMM result for annual gamete output (AGO) in function of "Period".**

|  | Fixed effects | Estimate coeff | Std. error | t-value | *p*-value |
|---|---|---|---|---|---|
| AGO | Intercept | 0.049250 | 0.006017 | 8.185 | 9.63e-06 |
|  | Period 2 | −0.012300 | 0.008510 | −1.445 | 0.17894 |
|  | Period 3 | −0.017400 | 0.008510 | −2.045 | 0.06811 |
|  | Period 4 | 0.027900 | 0.008510 | 3.279 | **0.00831** |
|  | Period 5 | 0.006900 | 0.008510 | 0.811 | 0.43633 |
|  | **Random effect** | **Variance** | **Std. dev** |  |  |
|  | Size | 7.241e-05 | 0.00851 |  |  |
| LogLink = 33.5; R$^2$ = 0.79 |  |  |  |  |  |

**Note:**
The "Size", was set as random effect. Estimate coefficient, Standard Error, t-value and significance level (*p*-value) are provided for fixed effects. Values in bold are statistically significant.

also found in March and was 6.3 ± 0.5% and 5.41 ± 0.61% for CS and US, respectively (Fig. 2 and Table 2). Conversely, in the harvest locality, the average IGO value increased over the years and the higher AGO was estimated during the last periods, when values were 0.13 and 0.11 g g$^{-1}$ yr$^{-1}$, respectively for P4 (*p*-value = 0.008) and P5 (Fig. 3 and Table 3). During P5 in the protected locality, the average IGO was 0.026 g g$^{-1}$ se$^{-1}$ and the AGO was 0.11 g g$^{-1}$ yr$^{-1}$ (Table 2).

## Population density and reproductive potential

In the harvest locality the density of CS did not change significantly from P1 (0.6 ± 0.2 ind m$^{-2}$) to P5 (0.5 ± 0.1 ind m$^{-2}$), while the US density decreased significantly from 2.7 ± 0.3 to1.2 ± 0.4 ind m$^{-2}$ over the same years (Fig. 4, Table 4, Fig. S4). Variations in density of recruits, juveniles and middle size-class individuals were also non-significant between P1 and P5 (Fig. 4, Table 4 and Fig. S4).

Accordingly, in the harvest locality, the reproductive potential of CS was 0.03 during P1 and 0.04 g g$^{-1}$ m$^{-2}$ yr$^{-1}$ during P5, whilst in the same periods was 0.12 and 0.05 g g$^{-1}$ m$^{-2}$

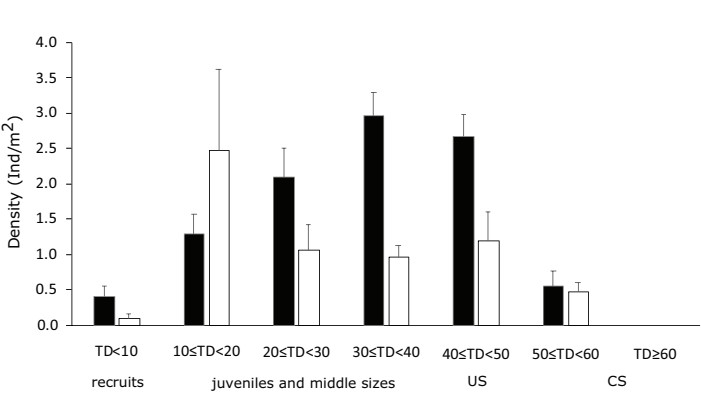
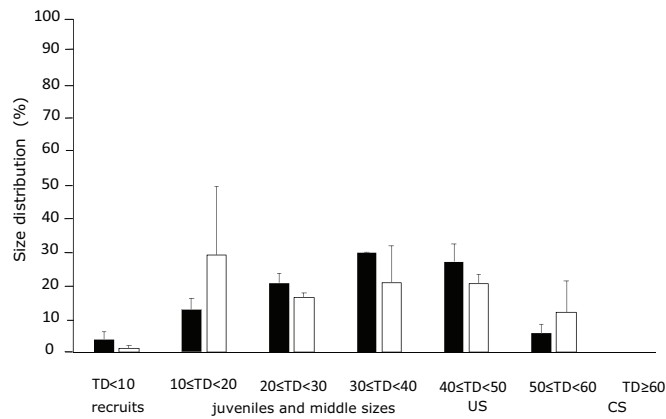

**Figure 4 Sea urchin population structure.** Size-density and size frequency distributions of the sea urchin population in the harvest locality during P1 and P5. The range of the size classes is 10 mm of test diameter without spines (TD). Size-classes are grouped by stage in relation with the main ecological and anthropogenic processes influencing them (recruitment, predation and human harvesting).

**Table 4 LMM result of changes in population demographic structure in the harvest locality between P1 and P5.**

| (a) | Fixed effects | Estimate coeff | Std. error | t-value | p-value |
|---|---|---|---|---|---|
| US density | Intercept | 2.7500 | 0.6080 | 4.523 | 0.0409 |
| | Period | −1.4167 | 0.5765 | −2.457 | **0.0436** |
| | Random effect | Variance | Std. dev | | |
| | Site | 0.3405 | 0.5835 | | |
| AIC = 33.21; R² = 0.52 | | | | | |
| (b) | Fixed effects | Estimate coeff | Std. error | t-value | p-value |
| CS density | Intercept | 0.20621 | 0.09052 | 2.278 | 0.0352 |
| | Period | −0.02516 | 0.11687 | -0.215 | 0.8320 |
| | Random effect | Variance | Std. dev | | |
| | Site | 0 | 0 | | |
| AIC = 23.3; R² = 0.01 | | | | | |
| (c) | Fixed effects | Estimate coeff | Std. error | t-value | p-value |
| Mid size density | Intercept | 1.0919 | 0.1951 | 5.597 | 0.0326 |
| | Period | −0.3490 | 0.1733 | −2.013 | 0.0541 |
| | Random effect | Variance | Std. dev | | |
| | Site | 0.02326 | 0.4823 | | |
| AIC = 51.29; R² = 0.24 | | | | | |
| (d) | Fixed effects | Estimate coeff | Std. error | t-value | p-value |
| Recruits | Intercept | 0.29574 | 0.09478 | 3.12 | 0.0142 |
| | Period | -0.20927 | 0.12236 | -1.71 | 0.1256 |
| | Random effect | Variance | Std. dev | | |
| | Site | 0 | 0 | | |
| AIC = 7.27; R² = 0.01 | | | | | |

**Note:**
"Period" is set as fixed effect and "Site" is set as random effects. Estimate coefficient, Standard Error, t-value and significance level (p-value) are provided for fixed effect. Values in bold are statistically significant.

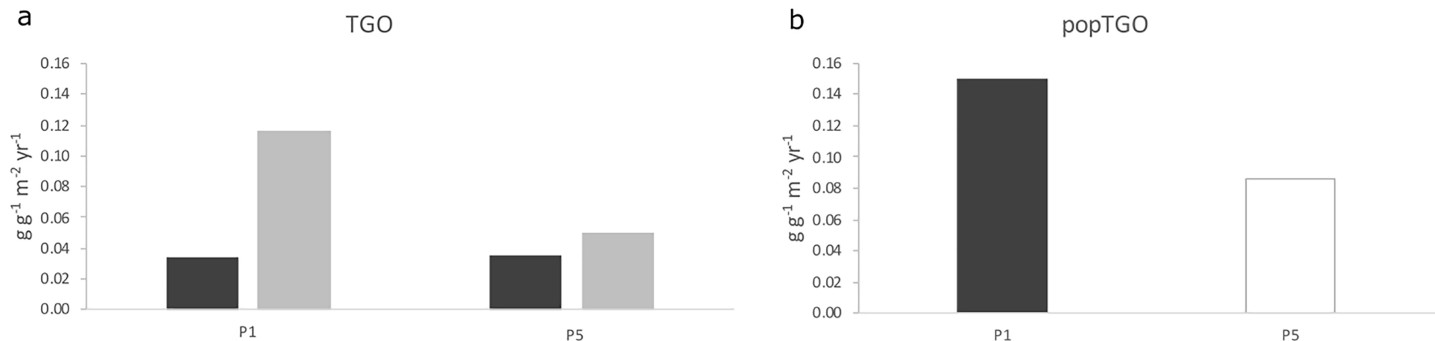

**Figure 5 Population reproductive potential.** Total gamete output (TGO, g g$^{-1}$ m$^{-2}$ yr$^{-1}$) of the CS and US size-classes and the reproductive potential of the whole population (popTGO) of the harvest locality during P1 and P5.

yr$^{-1}$ for US (Fig. 5 and Table 2). In the protected locality during P5, density of CS and US was 0.9 ± 0.4 and 0.8 ± 0.4 ind m$^{-2}$, respectively, and the population reproductive potential was 0.04 and 0.005 g g$^{-1}$ m$^{-2}$ yr$^{-1}$ for CS and US, respectively (Fig. 5, Table 2). Consistently with these results, the reproductive potential (TGO) of the harvested population was 0.15 during P1 and 0.086 g g$^{-1}$ m$^{-2}$ yr$^{-1}$ during P5 (Fig. 5, Table 2), whilst it was 0.09 g g$^{-1}$ m$^{-2}$ yr$^{-1}$ for the protected population during P5 (Table 2).

## DISCUSSION

The temporal pattern of the monthly mean GSI at the harvest locality was not significantly different over the years, but it was significantly higher in the large individuals (CS) rather than in the smaller ones (US), confirming the major role of commercial size urchins on the population breeding stock. The main spawning events occurred generally in spring according with *Spirlet, Grosjean & Jangoux (1998)*, except for P3 when there was no evidence of abrupt spawning events. Finally, during the last sampling period (P5), the monthly mean GSI of the population at the harvest locality was not significantly different from the protected locality.

Due to the overexploitation, density of CS urchins in the harvest locality remained at low values across the 5-year period (less than one individual per square meter). Contrarily, US density significantly decreased from more than two individuals to less than one individual per square meter, leading to a proportional decrease of their gamete output and the loss of 40% of the whole population reproductive potential.

In the North-Western Sardinia, the intensive size-selective sea urchin harvest strongly affects the density of the largest (CS) individuals (*Pais et al., 2007*; *Ceccherelli et al., 2022*). Su Pallosu Bay is one of the main hotspots along this coast, where the harvest of sea urchins determines the persistence of truncated demographic population structure above the commercial size-class over time (*Loi et al., 2017*). To date, the scarcity of the major population breeding stock (CS) was buffered by the high natural density of younger animals (US) and supported by an effective recruitment rate in the area (*Farina et al., 2018*).

Nevertheless, our results suggest that the intensive harvest in recent years has drastically rescaled the population body size, often exceeding in the legal limit of commercial size and

thus affecting the youngest breeding stock, as demonstrated by the density of US individuals almost halved in 5 years. Although no data are available to estimate larval connectivity among the harvest population and outside populations, our results suggest that the sea urchin recruitment in the future years will have to totally rely on larvae coming from other populations.

The density of the commercial size-class in the protected locality was estimated during the last period of the study (P5) and corresponded to an intermediate value with respect to the density recorded before (2010) and after (2012) the stock contraction caused by the intensive harvesting practiced in the past (see *Coppa et al., 2021*). In effect, stricter sea urchin fishery regulations were introduced in the MPA after the population contraction (*Marra et al., 2016*; *Pieraccini, Coppa & de Lucia, 2016*). Accordingly, despite the GSI was no estimated during P1 for the protected population, a slight recovery of the commercial size-class after the collapse kept stable the reproductive potential of the population over the 5-year monitoring period. Conversely, the continuous and inexorable deterioration of the reproductive potential of the harvest population could be a direct consequence of the harvesting activity aiming to market the gonads of individuals under commercial legal size (*Furesi et al., 2016*).

However, indirect effects can be manifested also at community level (*Kaiser & Jennings, 2001*), since the key role of the functional herbivore *P. lividus* in the ecosystems (*Boudouresque & Verlaque, 2001*). For example, the intensive harvesting of *P. lividus* can encourage the proliferation of the habitat competitor *Arbacia Lixula*, a sea urchin species harvested by humans and weakly preyed in nature (*Guidetti, 2004*).

Moreover, the rescaling population body-size can also impair the fertility, in terms of gamete quality, egg size (*Moran & McAlister, 2009*), larval development and survival (*Berkeley, Chapman & Sogard, 2004*), and it pushes the smallest fertile sea urchins to increase their reproductive investment (*Fenberg & Roy, 2008*). Consistently with these mechanisms, our results seem to suggest a growing trend of the IGO across the years, indicating the increase over time of the amount of gamete per urchin per spawning event. Accordingly, the AGO was high in the last 2 years, especially during P4.

Nevertheless, the reproductive investment is a well-documented compensatory response that induces physiological and behavioural changes in the survivors, such as alterations in the timing of maturation, metabolism and growth rate (*Ali, Nicieza & Wootton, 2003*; *Enberg et al., 2012*). Although a long time series of data need to confirm this trend, it would wonder whether the persistence of the harvesting pressure could induce survivors to improve their reproductive investment in gonad production rate or spawning intensity to deal with the chronic lack of the major breeding stock (largest commercial size-class urchins).

## CONCLUSIONS

The intensive size-selective harvest has deeply changed the demographic structure of the sea urchin population and has determined a drastic decrease of its reproductive potential, by affecting the portion of the youngest part of the breeding stock. An in-depth analysis of

the results also highlights an interesting growing trend of the amount of gamete emitted by the survivors.

Due to such evidence, mitigating the ecological consequences of size-selective sea urchin exploitation in this area probably requires a shift in management strategies designed to modulate yields on the natural variability in size-classes characterizing the sea urchin population structure. Ideally, it would be useful to address local harvest activities towards a size-balanced population fishing, avoiding the complete depletion of the largest classes (*e.g.*, *Law & Plank, 2018*). Such approach should necessarily go through a shared vision with the stakeholders that includes the rise of awareness of fishermen and much more effective control of illegal practices. Actually, in the context of the small-scale fisheries, the engagement of local fisheries and other stakeholders to establishing a co-management mechanism of the resources would improve achieving an effective fishery governance (*e.g.*, *Mahon et al., 2003*; *Sousa et al., 2020*). The effects of the rescaling sea urchin population body-size identified in this study can contribute to the development of an ecosystem-based-management fishing plan to ensure the sustainable exploitation of the resource and the conservation of the ecosystem.

## ACKNOWLEDGEMENTS

The authors would like to thank the administration of the Marine Protected Area "Penisola del Sinis-Isola di Mal di Ventre" for providing the boat for the monthly sampling and all the students who helped in the field and in the laboratory during these years.

### Funding

This study was supported by the Autonomous Region of Sardinia (RAS) with grants from L.R. 7/2007, L.R. 37/1998, and L.R. 20/2015. Simone Farina and Giulia Ceccherelli received support from the NBFC to SZN Institute and University of Sassari, funded by the Italian Ministry of University and Research, PNRR, Missione 4 Comporeceived nente 2, "Dalla ricerca all'impresa", Investimento 1.4, Project CN00000033. The funders had no role in study design, data collection and analysis, decision to publish, or preparation of the manuscript.

### Grant Disclosures

The following grant information was disclosed by the authors:
Autonomous Region of Sardinia (RAS): L.R. 7/2007, L.R. 37/1998, and L.R. 20/2015.
NBFC to SZN Institute and University of Sassari.
Italian Ministry of University and Research: CN00000033.

### Competing Interests

Gianni Brundu, Daniele Grech and Barbara Loi are employed by International Marine Centre. Ivan Guala is a project-based contracted by International Marine Centre.

## Author Contributions

- Nicole Ruberti performed the experiments, analyzed the data, prepared figures and/or tables, authored or reviewed drafts of the article, and approved the final draft.
- Gianni Brundu performed the experiments, authored or reviewed drafts of the article, and approved the final draft.
- Giulia Ceccherelli analyzed the data, authored or reviewed drafts of the article, and approved the final draft.
- Daniele Grech performed the experiments, authored or reviewed drafts of the article, and approved the final draft.
- Ivan Guala performed the experiments, authored or reviewed drafts of the article, and approved the final draft.
- Barbara Loi performed the experiments, authored or reviewed drafts of the article, and approved the final draft.
- Simone Farina conceived and designed the experiments, performed the experiments, analyzed the data, prepared figures and/or tables, authored or reviewed drafts of the article, and approved the final draft.

## Data Availability

The raw data is available in the Supplemental Files.

## Supplemental Information

Supplemental information for this article can be found online at http://dx.doi.org/10.7717/peerj.16220#supplemental-information.

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
