# Peer review of "Intensive sea urchin harvest rescales Paracentrotus lividus population structure and threatens self-sustenance"

_PeerJ, doi:10.7717/peerj.16220_

## Round 0.1 · original submission · Major Revisions

This is an interesting study that explores the potential and the reproductive cycle of P. lividus in Sardinia where strong exploitation of the resource is ongoing and guidelines for the rational management of natural stocks are needed. The research was carried out on a very extended time scale and therefore could be useful for dynamics understanding.

All three reviewers agree on the importance of your work and that with this publication you are filling a knowledge gap. Nerveless there are some points raised by the reviewers which must be addressed. I invite you to resubmit your manuscript after addressing all reviewer comments.

·

Basic reporting

The article is interesting and was carried out over a very long period of time, however it has some weaknesses in the description of the methodology and in interruption of sampling that should be explained and justified somehow in the methodology.
Line 193 – complete the reference Ourens
Line 232 – H or C?
Line 260 – insert units and accuracy of size and weight and give more details about the size measurement
Line 263 – insert units in the formula
Line 270-274 – include references, this methodology was used by other authors?
Line 289-291 – this definition of TGO was used by other authors, some references?
Line 327 – 350 – The temperature data should also be analysed inside each year and correlated with the GSI so it can be with the results of the next subchapter. The GSI is not only influenced by size but also by monthly temperature. I think the author should revise this part of the results with the temperature data of the months mentioned in the text.

Experimental design

Methods should include references of the methodology used in some cases, as TGO, IGO, AGO...
The author should also justify in the methodology the lack of data in some months, fig 2.

Validity of the findings

no comment

Reviewer 2 ·

Basic reporting

This study explored the reproductive cycle and potential of an ecologically and commercially important sea urchin, P. lividus. The manuscript is generally well-written, and the supporting literature in the introduction is excellent. Nonetheless, I have some serious concerns about the analytical approaches used to support the authors’ claims. Additionally, I suggest that the authors explicitly address, wherever relevant, all possible alternative explanations for the observed differences in size structure and GSI, such as differences in refuge availability between sites (i.e., sites with greater refuge space may hold more urchins), potential variation in recruitment success among sites, and especially resource availability (which I expand on below). This could take the form of either an added sub-section within the Discussion, or brief explanations throughout the current Discussion text. Moreover, the authors could include more detailed analyses of size structure to further support their claims. Finally, the final paragraph of the intro includes some expectations, but the testable hypotheses could be more explicitly stated in both the intro and the methods.

Experimental design

I have a serious concern about the utility of the MPA as a reference site for GSI. Because the MPA was used as a reference in the study design, it should be comparable in terms of physical features (substrate type and complexity) and especially resource availability. Is it possible to provide any context about the physical attributes of the MPA vs the harvest sites? Without demonstrating that resource availability (i.e., macroalgae cover) was contextually similar between the open-harvest sites and the MPA, it is impossible to evaluate whether the differences in GSI and size structure are attributable to selective harvest. If these data are not available, then the authors should at a minimum clearly state these limitations and alternative hypotheses in the discussion.

If the authors can address my concern above, I would like to see an additional analysis using a similar GLM model, but using a response ratio approach. Currently, the GLMM uses site location as a random effect in the model. The response ratio would be calculated as the log(GSI in C locality / GSI H locality) for the years where sampling was conducted at both locations. The predictor variables would remain the same. The advantage of using the response ratio is to look at proportional changes.

I also believe an analysis of annual size frequency distribution is needed to further support the authors’ claims. Figure 4 only shows urchin size structure at the H locality pooled across years. What was the size structure at the C locality? I checked the supplementary material and cannot find this information. A figure that depicts annual size frequency at both locations would be helpful. There is also no quantitative analysis of the size of the size distribution. It would be helpful to see a quantitative measure of skewness of overall mean between each of the sites.

Validity of the findings

The effect of size-selective commercial harvest on the population dynamics of a dominant benthic mesopredator is an important topic. However, I believe the concerns raised above need to be thoroughly addressed before the manuscript is reconsidered.

Lines 391-392: “The systematic removal of the largest sea urchins at the H locality has determined a truncated demographic structure ..” It is my opinion that the authors have not adequately provided support for this claim. I checked the supplementary material and did not find an annual size-frequency distribution plot for each site. Fig 4 only shows the averages, but an annual plot is needed to support the claim.

Additional comments

Introduction:

I found the introduction unnecessarily long. The authors might consider combining and reducing some of the text between lines 144 and 158. There is some redundancy in the text between lines 159 and 194 that could also be condensed to shorten the intro.

Line 135: These shifts to barrens can also have substantial impacts on the GSI and harvestable condition of sea urchins.

The final paragraph of the introduction includes some expectations, but the authors should explicitly state their hypotheses. This should be reflected in the methods as well.


Methods:

Each hypothesis should be clearly stated. e.g., “X analysis was used to test the hypothesis that … “

Line 250: The terminology “H locality” and “C locality” is somewhat confusing. Can these just be called “harvest” and “control” throughout?

Results:
I suggest the authors lead the results section with the most interesting result instead of the environmental conditions.

Line 339: change to “..locality resulted in a non-significant response when…”

Line 383: this is the most interesting result.

Discussion:
Lines 410-411: this result should be supported with a figure in the results section. Figure 4 does not capture this.


Figures:

Figure 2. Rather than faceting by sampling period, I’d like to see a single plot facetted by site type (H or C) each with a line for average CS, average US, and temperature, with error bars depicting the propagated variability across sampled years.

Figure 3. What do the error bars represent? Also, what is the purpose of what I assume is a dotted regression line? The regression assumes that changes are linear and predicted, but the ms suggests that GSI peaks in P3 and is therefore non-linear. Instead, it would be helpful to see the results of the ANOVA between periods with asterisks above those that are significantly different. Additionally, I strongly suggest adding a third column that is the response ratio between H and C for the P5 period, as I described in my comment above. Please also include error for the response ratio to demonstrate whether it is significant from 0 (i.e., if positive, then higher GSI in the Control, if negative, then higher GSI in the H).

Figure 4. What are the error intervals on each bar? Additionally, there were very few commercial-sized urchins observed in either the P1 or P5 periods. The authors should include an annual size frequency plot. Moreover, the density differences are quite different between size classes (over for 20-30, 30-40, 40-50) and I wonder how much of this is driven by annual variability?

·

Basic reporting

The manuscript is well organized, with state-of-the-art well documented and the objectives well described. In terms of language use, I think that the text can be improved to ensure a better understanding of methodology application and clarity of the results obtained. I suggest you have a colleague who is proficient in English and familiar with the subject matter to review your manuscript. In the commented file, I make some indications of sentences that can be rephrased to improve clarity of the text meaning.

Experimental design

The research questions is well defined under the light of the sate-of-art presented and initial evidences provided. Nevertheless, I have some reserves about the methodological approach. The authors support their analysis of the reproductive potential of local populations of sea urchins on the variation of total gonad weight before and after spawning. In my perspective this is only an indirect form of evaluate spawning and fecundity, and this must be highlighted. The sea urchin gonads are composed of both nutritive phagocytes (somatic tissue) and gametes, which abundance and distribution changes across the gametogenesis. I think that it is worthwhile to mention the rationale behind this methodological approach hopefully including some information about the correlation between gonad size and fecundity in that population. It is noteworthy that nutritional condition of sea urchins, number of spawning events and maturity level of gonads (to which also exists evidences of partial spawning, Byrne et al., 1990) affect the egg size and the fecundity (Meidel & Scheibling, 1999; O’Hara & Thórarinsdóttir, 2021).
Regarding the determination of IGO and AGO, I think that it is worthwhile to actual indicate the equations that are beyond the determination of mean individual gamete output and annual gamete output, I had difficulties to understand the rationale behind the units used to represent the gamete output. Furthermore, the identification of single spawning events based in the decrease of GSI requires validation based in the analysis of gonads maturity of sampled urchins. While abrupt changes on GSI can be indicative of massive spawning of local populations, changes more subtile can be more difficult to support as they can also result from sampling bias.
In lines 264-265, you assume a balanced sex ratio for the population sampled, but I think you need to present evidence of this equilibrium between males and females in your samples. I checked in the reference paper Loi et al (2017) and no information on sex ratio was provided there. Also, do you have evidence of no differences between females and males regarding GSI and spawning peak? I think this is important to include here to make your results more robust.
Regarding the statistical analysis of environmental data, I recall that monthly SST (as other seasonal variables) are normally considered depended observations and for that reason analysis of variance is not the best method to identify differences between observations. This data set must be detrend before any analysis and then analysed using, for instances, autocorrelation and cross-correlation methods. Nevertheless, in the case of this study, as seasonality appears to be the most important factor of influence, maybe it could not be necessary to provide such analysis.
I have other specific comments regarding the statistical analysis in the commented file uploaded.

Bibliographic references
Meidel, S. K., & Scheibling, R. E. (1999). Effects of food type and ration on reproductive maturation and growth of the sea urchin Strongylocentrotus droebachiensis. Marine Biology, 134, 155-166.
O’Hara, T. E., & Thórarinsdóttir, G. G. (2021). A depth-dependent assessment of annual variability in gonad index, reproductive cycle (gametogenesis) and roe quality of the green sea urchin (Strongylocentrotus droebachiensis) in Breidafjördur, west Iceland. Regional Studies in Marine Science, 45, 101846.

Validity of the findings

Based in my comments to the experimental design, I think that results and the findings described in the manuscript need some level of validity. In my opinion it is missing a histological analysis of sea urchins to support the spawning events identified. Probably, the extension of Brewin et al (2000) methodology to identify single/individual spawning events is a bit abusive and you could limit your analysis to just Annual spawning and include the determination of spawning magnitude. I missed for summary table of the sampling made including the number of animals sampled in each event by size class. Looking for figure 2 it is evident that the sampling effort (even that weather conditions related) was very irregular across the study period, and I find very difficult to identify all the spawning events presented in Table S3 , which I think it should be included in the main text. The tables presented as supplementary material clear need legend and units of variables described to facilitated the interpretation of the information given.
Based in the results obtained is not clear to me that the changes observed in population structure in this region of Sardinia is due to changes in the population fecundity and fertilization success and not just because smaller urchins are becoming more and more harvest as the larger ones disappear. To take conclusions of this intensification of harvesting across smaller size classes, I think you need to include more direct measures of maturation and fecundity and also sea urchins age – how old are these smaller urchins? How many opportunities they had to contribute to the settlers’ pool? I think that the answer to these questions would contribute to the evidence of population truncation in smaller size-classes.
I found that the conclusions take were a bit speculative under the light of the evidence that you were able to confirm. I make some detailed comments in the uploaded file.

---

## Round 0.2 · Minor Revisions

I have assessed the revision myself, and I'm satisfied with the second version of this manuscript since the authors have addressed all the previous reviewers' comments.

From my point of view, the manuscript is almost ready for publication.

I ask the authors to address the suggestions raised by reviewer 2 and to return it back

Reviewer 2 ·

Basic reporting

The manuscript is clear and provides specific objectives that are appropriately and thoroughly addressed.

Experimental design

The authors did an excellent job of addressing my initial analytical concerns.

Validity of the findings

The findings are appropriate and clearly stated.

Additional comments

I commend the authors for making substantial revisions that thoroughly addressed all of my concerns. I believe the manuscript is suitable for publication.

·

Basic reporting

The effort made by the authors to meet the concerns and suggestions made by the reviewers is very valuable. At this stage, I think that the manuscript needs a throughout revision to improve the use of English language. Particularly, some of the phrasing used in the M&M and results section is difficult to follow and, could certainly improve with a careful revision of the language use will make the message more clear. The new figures proposed – Figure 3 and Figure 4 – still need some work to be accepted for publish (Please check some detail comments to the V1).

Experimental design

I acknowledge the effort conducted by the authors to improve the analysis conducted based on the previously knowledge on the study area and the data collected by them on the local urchin population. At this stage, some of the information provided in the M&M needs to improved in terms of language use. I provide some suggestions in the text.

Validity of the findings

I believe the re-analysis conducted by the authors provide a better understanding and validation of their findings. I do not have much to add to this new version. Some small comments can be find in the commented version provided

---

## Round 0.3 · accepted · Accept

The authors have addressed all of the reviewers' comments.
This manuscript is ready for publication.